# Down-Regulation of Strigolactone Biosynthesis Gene *D17* Alters the VOC Content and Increases *Sogatella furcifera* Infectivity in Rice

Shanshan Li [1], Hualiang He [1], Lin Qiu [1], Qiao Gao [1], Youzhi Li [1,2] and Wenbing Ding [1,2,*]

1 College of Plant Protection, Hunan Agricultural University, Changsha 410128, China
2 Hunan Provincial Engineering & Technology Research Center for Biopesticide and Formulation Processing, Changsha 410128, China
* Correspondence: dingwenb119@hunau.edu.cn

**Abstract:** *DWARF17* (*D17/HTD1*) is a well-defined rice strigolactone (SL) biosynthesis gene that influences rice tiller development and the production of rice. To investigate whether SLs play a role in the regulation of rice's defense against the white-backed planthopper (WBPH, *Sogatella furcifera*), we compared a SL-biosynthetic defective mutant (*d17*) with WT rice plants. Our olfactory bioassay results revealed that WBPHs are attracted to *d17* plants, which may be attributed to changes in rice volatile substances. Hexanal, a volatile substance, was significantly reduced in the *d17* plants, and it was demonstrated that it repelled WBPHs at a concentration of 100 µL/L. Compared to the WT plants, WBPH female adults preferred to oviposit on *d17* plants, where the egg hatching rate was higher. The transcript level analysis of defense-associated genes in the JA and SA pathways showed that the expression of *OsJAmyb*, *OsJAZ8*, *OsPR1a* and *OsWRKY62* were significantly reduced in *d17* plants compared to WT plants following WBPH infection. These findings suggest that silencing the strigolactone biosynthesis gene *D17* weakens defenses against *S. furcifera* in rice.

**Keywords:** *DWARF17*; *D17*; strigolactones; *Sogatella furcifera*; rice defense

## 1. Introduction

Rice is one of the most important grain crops in the world, and it is severely affected by a variety of insect pests [1]. One of the most important insect pests, white-backed planthopper (WBPH, *Sogatella furcifera*), feeds on the sap of rice plants, hindering nutrient delivery and causing damaged plants to become yellow and dry [2,3]. Rice withstands herbivorous insect damage through defense mechanisms, such as the accumulation of toxic substances, the formation of physical barriers and releasing volatile organic compounds (VOCs) [4–7]. VOCs such as (Z)-3-hexenal, (Z)-3-hexen-1-ol, (E)-2-hexanal and others are thought to influence insect host selection behavior and reproductive behavior [8–10]. Additionally, insect feeding activates the signal pathways of the plant hormones jasmonic acid (JA) and salicylic acid (SA) in rice, resulting in the accumulation of secondary metabolites, and ultimately leading to direct and indirect pest resistance [11–15].

Strigolactone (SL) is a small terpenoid molecule recently identified as a plant hormone that can regulate plant development and physiological processes, which include seed germination, secondary growth, root development, branching and leaf senescence [16,17]. In addition, SLs also actively mediate abiotic and biotic stresses, such as drought stress, salinity stress, nutrient stress, pest and pathogen infection [18,19]. The SL biosynthetic pathway of rice consists of *DWARF27* (*D27*) [20], *DWARF17* (*D17*) [21] and *DWARF10* (*D10*) [22], while *DWARF14* (*D14*), *DWARF3* (*D3*) and *DWARF53* (*D53*) participate in SL signal perception [23,24]. Among them, *D17*, which encodes CAROTENOID CLEAVAGE DIOXYGENASE 7 (CCD7), controls a key step in SL biosynthesis [25]. *d17* mutant plants created by knocking out *D17* with CRISPR-Cas9 technology exhibited dwarfing and increased

tillering, which is caused by reducing each internode and panicle [21,26]. Furthermore, *D17* has been linked to biotic and abiotic stresses. For example, *max3*, which encodes the CCD7 protein in *Arabidopsis* mutant lines, has higher leaf stomatal density and stomatal closure delaying in response to drought and salt stress [27]. Moreover, when infected with *Fusarium oxysporum*, *Irpex* sp. and *Sclerotinia sclerotiorum*, the knockdown of *PpCCD7* in *Physcomitrella patens* mutants resulted in earlier and more severe disease symptoms compared to WT plants [28]. Additionally, *ccd7* plants (*Nicotiana attenuata*) are more vulnerable to specialist weevil (*Trichobaris mucorea*) larvae attack, producing more larvae that are also significantly larger [29]. However, whether *D17* participates in WBPH resistance remains unknown.

This study aimed to determine whether *D17* is involved in WBPH resistance in rice. To achieve this, we used well-defined *d17* defective rice mutants to conduct WBPH bioassays to determine differences in host selection behavior, the number of eggs laid, hatching and survival rates, changes in VOCs and insect-resistance genes between *d17* and WT plants. Our findings provide a new reference for the role of *D17* in mediating biotic stress tolerance.

## 2. Materials and Methods

### 2.1. Plant Growth and Insects

The rice genotype Nipponbare (WT) and *d17* mutant plants came from Dr. Zeng Dali, China National Rice Research Institute, Zhejiang, China [23]. All rice plants were cultivated in a greenhouse kept at 28–32 °C, 80 ± 5% relative humidity and exposed to natural sunlight. The rice seeds, which were soaked well in water, were grown for 10 days then transplanted into 10 L of Kimura B nutrient salts (Coolaber, Beijing, China) for 30 d, 45 d and 60 d to use for experiments, respectively.

WBPHs were collected from a rice field in Changsha, Hunan, China, and fed on rice seedlings at 26 °C in an artificial climate incubator. The WBPHs were used for experiments after propagating three generations continuously.

### 2.2. Y-Tube Olfactometer Bioassay

The selection preferences of WBPHs between the *d17* and WT plants from the 30 d, 45 d and 60 d rice growing sites were compared using a Y-tube olfactometer bioassay (Figure 1a), according to the method described by Wang et al. and Zhao et al. [30,31]. In each experiment, an individual WBPH was starved for an hour then placed in the straight arm of the Y-tube, and the time was recorded till it arrived at the junction of the Y-tube. A WBPH was considered to have made a successful selection when it crawled into an arm, reached a position 3.5 cm away from the Y-junction and stayed in this position for 1 min. Otherwise, it was recorded as a failure if it failed to make a choice within 10 min [30]. The left and right arms of the Y-tube were switched after each test, and it was removed after every two tests to clean it. The host selection test was conducted in a dark case with the temperature maintained at 26 ± 1 °C and RH = 50%. Three biological replicates were tested, with the selection preference of 15 WBPHs recorded in each repeat.

For the determination of the olfactory behavioral response of WBPH to VOCs in rice (Figure 1b), we dissolved the standard compound sample in chromatographically pure n-hexane (Macklin, Rochelle, IL, USA) at a concentration of 100 µL/L and added 100 µL drops at a time to a filter paper strip (1 × 10 cm$^2$). The blank control was 100 µL n-Hexane.

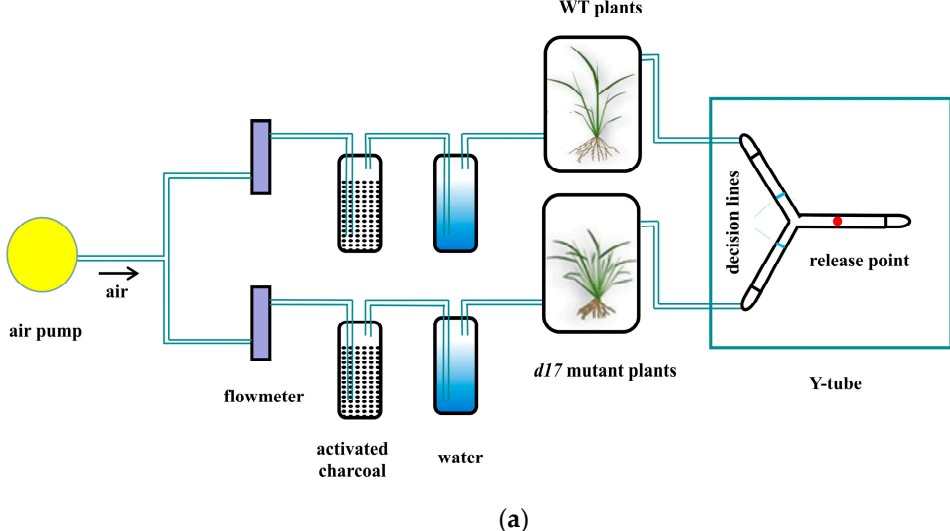

(**a**)

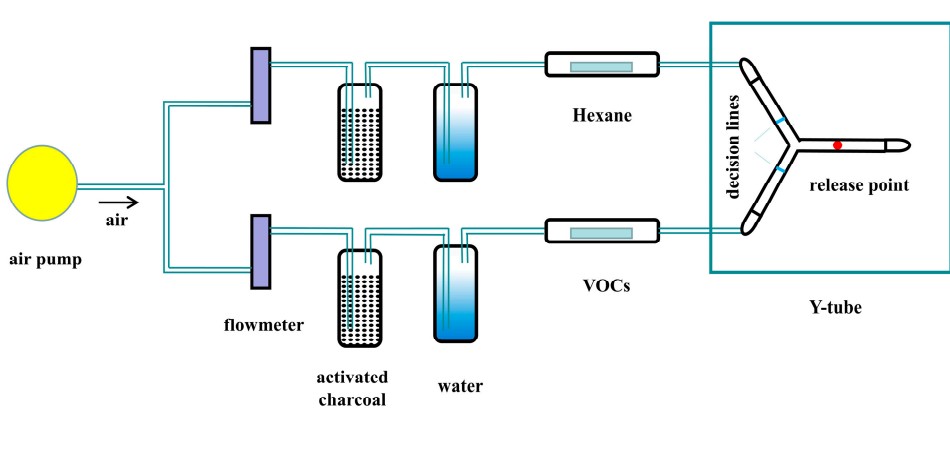

(**b**)

**Figure 1.** (**a**,**b**) Schematic diagram for Y-tube bioassay. (**a**) *d17* mutant plants and WT plants used for pairwise comparison were placed into the two odor source bottles. (**b**) Hexane and VOCs used for pairwise comparison were placed into the two odor source bottles. Air was forced through the air pump, flowmeter, activated charcoal, water, odor source bottles and Y-tube in that order. The air was purified using active carbon and water, respectively. The air flow in each arm of the Y-tube was kept at 300 mL/min, and the host selection test was conducted in a dark case.

### 2.3. Collection and Identification of Rice VOCs

The VOCs from the *d17* and WT plants were collected separately using Tenax-TA (60~80 mesh, 100 mg) from 9:00 to 17:00 each day. Following 8 h of collection, 800 μL of n-hexane (Macklin, Rochelle, IL, USA) was used to rinse the adsorbent. The elution was then placed in a 1.5 mL brown sample bottle and kept at −80 °C for storage. The test was repeated at least 3 times and three rice plants were taken as a sample for the *d17* mutant plants or WT plants.

The VOCs were analyzed using a GCMS-QP2010 Ultra (Shimadzu, Columbia, MD, USA) instrument. Each sample was injected in a volume of 1 μL with a split injector kept at 230 °C. The flow rate was maintained at 1 mL/min by transporting He. The oven temperature was first held at 40 °C for 2 min, and then increased to 250 °C at 6 °C/min. Electronic impact (EI) spectra were recorded at 70 eV in scan mode from 33 to 300 amu.

### 2.4. WBPH Bioassays

To investigate the difference in the oviposition selection of WBPHs, 10 gravid WBPH females were placed in a pot with two 30 d plants (a *d17* plant and a WT plant) for 24 h. After 24 h, we released the WBPHs and counted the number of eggs and ovipositing marks under a microscope. This experiment was repeated at least 10 times for each plant pair.

In order to determine the difference in the hatching percentage of WBPHs, 10 gravid WBPH females were placed in a pot that had contained a 30 d plant (a *d17* plant or a WT plant). We removed WBPHs when they oviposited and colonized at 24 h, and subsequently recorded the number of hatched nymphs every day until there were no more nymphs emerging. After that, unhatched eggs were counted under the microscope. The experiment was repeated at least 10 times for each line.

To determine the difference in the survival rate of WBPHs, 30 nymphs of WBPHs were also placed in a flowerpot containing plants for 30 days (a *d17* plant or a WT plant). We counted the number of WBPHs every two days until the 8th day. The experiment was also repeated at least 10 times for each line.

### 2.5. WBPH Infestation

For this test, a 30 d plant was, respectively, damaged at 0 h, 6 h, 12 h and 24 h with ten 5th instar nymphs confined in a pot. The test was repeated three times and four rice plants were taken as a sample for each strain.

### 2.6. RNA Isolation and RT-qPCR

Total RNA was extracted from rice leaf sheath using RNAiso Plus (Takara, Tokyo, Japan). cDNA was synthesized using a PrimeScript™ RT reagent Kit with gDNA Eraser (Takara) according to the manufacturers' instructions. The RT-qPCR assay was carried out on a CFX96 Touch™ Real-Time System (Bio-Rad, Hercules, CA, USA) using the Hieff Unicon Universal TaqMan multiplex qPCR master mix (Yeasen, Shangai, China). The $2^{-\Delta\Delta Ct}$ method was used to analyze relative gene transcript levels [32]. The NCBI profile Server was used to design the RT-qPCR primers (http://www.ncbi.nlm.nih.gov/tools/primer-blast, accessed on 4 July 2022). The expression level of UBQ gene was used as an internal reference. A 5-fold dilution series of a bulked cDNA sample spanning five dilution points was used to establish a regression line to assess the amplification efficiency of the primer, and the amplification efficiency was regulated between 90% and 110%. The primers and probe sequences used for RT-qPCR analysis are listed in Table S1.

### 2.7. Statistical Analyses

Data are presented as mean values with standard deviation (SD). The Y-tube olfactometer bioassay and WBPH oviposition selection preference experiment were analyzed statistically using the chi-squared test, while other treatment data were analyzed using Student's *t*-test. All the statistical analyses were conducted using SPSS version 26.0 [33].

## 3. Results

### 3.1. Selection Preferences of WBPHs between the d17 and WT Plants

Prior to the experiment, we cultured *d17* plants that had higher tillering than the WT plants and tended towards dwarfism (Figure 2a). At 30 d, the WBPHs showed an extremely significant tendency towards selecting the *d17* plant ($x^2$ = 17.010, *p* = 0.0000, Figure 2b). At 45 d, the preference towards *d17* plants was weaker but remained significant ($x^2$ = 3.998, *p* = 0.0455, Figure 2b). In contrast, at 60 d, there was no significant difference in WBPHs' preference between the *d17* and WT plants ($x^2$ = 1.709, *p* = 0.0874, Figure 2b). These findings demonstrate that WBPHs have a significant preference towards *d17* plants at the seedling stage (30–45 d).

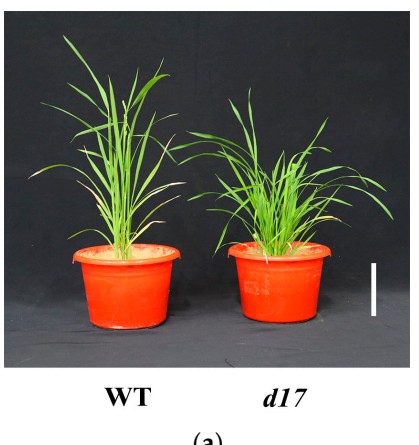

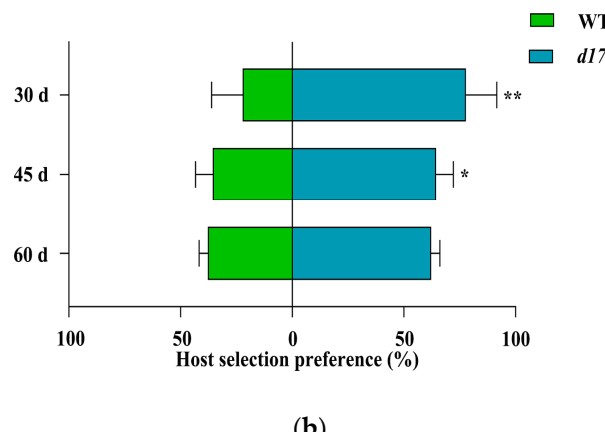

(**a**)

(**b**)

**Figure 2.** (**a**) Morphologies of the WT and *d17* rice plants at tiller stage. The scale bar is 10 cm. (**b**) Olfactory response of WBPHs to the WT and *d17* plants in the Y-tube olfactometer. Average (±SD) values from three biological replicates. Significant differences were calculated using the chi-squared test for dependent samples, * $p < 0.05$, ** $p < 0.01$.

### 3.2. Difference in Rice VOCs between the d17 and WT Plants

The volatile compounds of the *d17* and WT from the 30 d and 45 d rice growing sites were analyzed using GC-MS. Fourteen VOCs were identified from both the *d17* and WT plants: D-limone, hexanal, linalool, 2-nonen-ol, azulene, decanal, tetradecane, folic acid, cedrol, phytane, isopropyl myristate, dodecanol, phytol and squalene (Figure 3a,b). At 30 d, the WT plants emitted significantly more hexanal and decanal than the *d17* plants (t = 4.138, $p = 0.0144$; t= 3.240, $p = 0.0317$, Figure 3a). In contrast, the amount of linalool released from the *d17* plants decreased significantly at 45 d (t = 2.819, $p = 0.0479$, Figure 3b). These pure compounds were further evaluated using olfactory bioassays. At 100 µL/L, hexanal strongly repelled WBPHs ($x^2 = 7.490$, $p = 0.0062$, Figure 3c). In contrast, there was no significant effect of 100 µL/L decanal or linalool on WBPHs' selection preference. Thus, the change in VOCs caused by the *D17* mutation and the resulting reduction in hexanal may be an important reason why WBPHs are more attracted to *d17* plants.

### 3.3. Difference in WBPH Oviposition Selection and Hatching between the d17 and WT Plants

To further assess the adaptability of WBPHs to *d17* plants, we analyzed the differences in oviposition selection, hatching and WBPH viability between the *d17* and WT plants. Adult female WBPHs preferred to oviposit on *d17* plants, with 44.08% more WBPH eggs laid on *d17* plants than WT plants ($x^2 = 10.170$, $p = 0.0014$, Figure 4a). Similarly, there were 3.6 times more WBPH oviposition marks on the *d17* plants compared to WT plants ($x^2 = 18.360$, $p = 0.0000$, Figure 4b). Moreover, the WBPH egg hatching rate was 13.92% higher (t = 4.041, $p = 0.0008$, Figure 4c), but the developmental duration was not significantly different on *d17* plants compared to WT plants (t = 1.510, $p = 0.1310$, Figure 4d). Additionally, the survival rate of WBPHs was higher on *d17* plants, reaching a significant level on the eighth day (t = 2.417, $p = 0.0265$, Figure 4e). These findings indicate that the *D17* mutation favors the growth and reproduction of WBPHs.

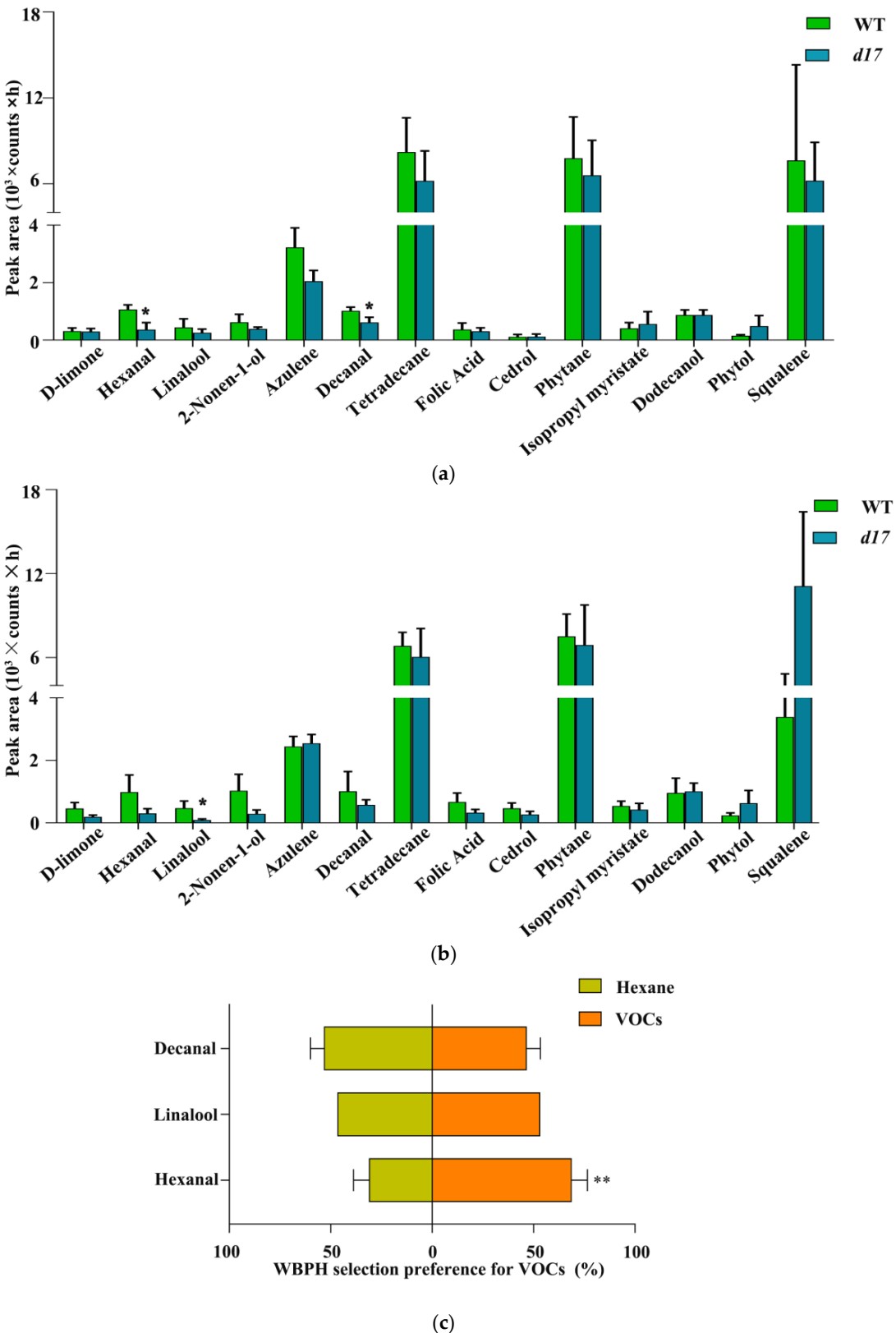

**Figure 3.** (**a**,**b**) Volatile organic compounds (VOCs) emitted from the WT and *d17* plants at 30 d and 45 d. Average (±SD) values from three biological replicates. Significant differences were determined using Student's *t*-test for dependent samples, * $p < 0.05$. (**c**) Olfactory response of WBPHs to VOCs in the Y-tube olfactometer. Average (±SD) values from three biological replicates. Significant differences were determined using the chi-squared test for dependent samples, ** $p < 0.01$.

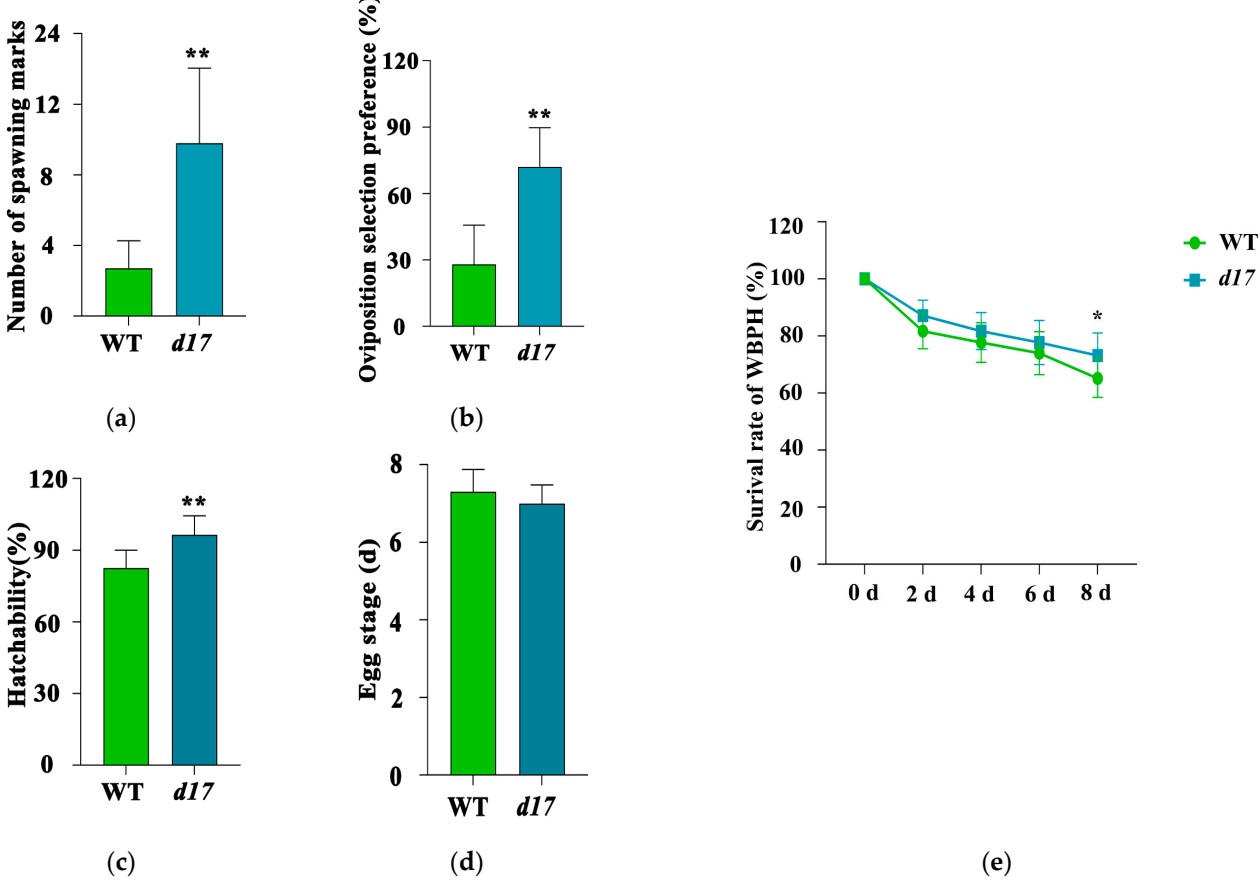

**Figure 4.** (**a**,**b**) Difference in WBPH oviposition selection preference between the WT and *d17* plants. Average (±SD) values from ten biological replicates. Significant differences were calculated using the chi-squared test for dependent samples, ** *p* < 0.01. (**c**) Difference in WBPH hatching rates between the WT and *d17* plants. Average (±SD) values from ten biological replicates. Significant differences were determined using Student's *t*-test for dependent samples, ** *p* < 0.01. (**d**) Difference in WBPH egg stage between the WT and *d17* plants. Average (±SD) values were 425 eggs from the WT plants and 386 eggs from the *d17* plants. Significant differences were determined using Student's *t*-test for dependent samples. (**e**) Difference in WBPH survival rate between the WT and *d17* plants. Average (±SD) values from ten biological replicates. Significant differences were determined using Student's *t*-test for dependent samples, * *p* < 0.05.

### 3.4. Response of Insect Resistance Genes in the d17 and WT Plants

Previous studies have suggested that the JA and SA signaling pathways aid in plant defense against phloem-sucking insects [12–15]. To further investigate whether JA and SA influence *d17* plant susceptibility to WBPHs, we measured the response changes in JA and SA signaling-associated genes in *d17* and WT plants following exposure to WBPHs for 6 h, 12 h and 24 h. We selected the JA response genes *OsJAZ8*, *OsJAMyb* and *OsPR10a*, and SA response genes *OsPR1a*, *OsWRKY45* and *OsWRKY62* [34–39]. Based on our RT-qPCR results, we found that WT plants infested with WBPHs for 6 h had significantly higher expression levels of the JA-associated gene *OsJAZ8* (t = 2.803, *p* = 0.0487, Figure 5a). At 0 h, 12 h and 24 h, *OsJAMyb* expression levels in WT plants were also significantly higher than in *d17* plants (t = 3.167, *p* = 0.0340; t = 2.850, *p* = 0.0464; t = 3.776, *p* = 0.0195, Figure 5b). In contrast, there was no significant change in the expression level of *OsPR10a* between the *d17* and WT plants (Figure 5d). WBPHs caused the expression levels of the SA-associated genes *OsPR1a* and *OsWRKY62* to be significantly higher in WT plants when infested with WBPHs for 6 h (t = 3.919, *p* = 0.0173, Figure 5c; t = 40.50, *p* = 0.0000, Figure 5f). At 24 h, the expression levels of *OsPR1a* in WT plants remained significantly higher compared to the *d17* plants at 24 h

(t = 7.074, *p* = 0.0021, Figure 5c), whereas there was no significant difference in *OsWRKY45* expression between the *d17* and WT plants (Figure 5e). Collectively, our results suggest that the *D17* mutation can down-regulate the expression of the JA signaling-associated genes *OsJAZ8* (6 h) and *OsJAMyb* (12 h, 24 h) and SA response genes *OsPR1a* (6 h, 24 h) and *OsWRKY62* (6 h) when the plant is infested with WBPHs.

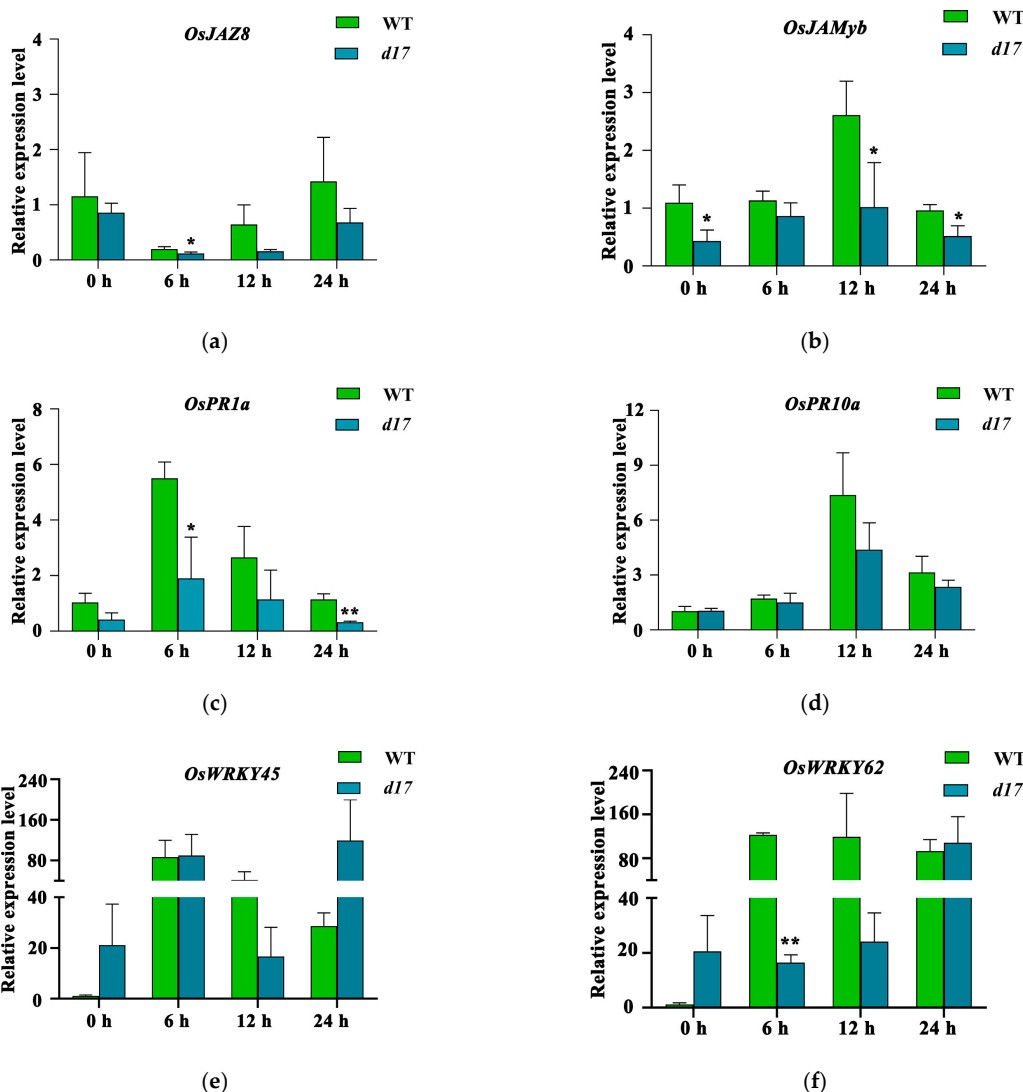

**Figure 5.** Changes in levels of expression of insect-resistant genes in the WT and *d17* plants following exposure to WBPHs for 0 h, 6 h, 12 h and 24 h. (**a**) The relative expression of *OsJAZ8.* (**b**) The relative expression of *OsJAMyb.* (**c**) The relative expression of *OsPR1a.* (**d**) The relative expression of *OsPR10a.* (**e**) The relative expression of *OsWRKY45.* (**f**) The relative expression of *OsWRKY62.* Average (±SD) values from three biological replicates. Significant differences were determined using Student's *t*-test for dependent samples, * $p < 0.05$, ** $p < 0.01$.

## 4. Discussion

In this study, we explored whether the strigolactone biosynthesis gene *D17* was involved in rice's defense against WBPH by comparing host selection behavior, oviposition, hatching and survival rate of WBPHs and the content of VOCs, and WBPHs induced different expression of the JA response genes and SA response genes between mutant *D17* (*d17*) and WT plants. Our results showed that the *d17* plants were more vulnerable to WBPHs.

We found that the WBPHs were more attracted to *d17* the plants than WT plants at days 30 and 45. As has been established, plant VOCs play an essential role in the host selection and oviposition behavior [40]. Previous research found that linalool strongly repelled brown rice planthoppers (BPH, *Nilaparvata lugens*) and that rice resistance against BPHs in the field decreases when linalool synthesis genes are silenced [34]. Furthermore, hexanal was found to inhibit *Phthorimaea operculella* larvae growth and adult oviposition, while decanal was found to inhibit *Cameraria ohridella* ovipositing [41,42]. Furthermore, the *D17* gene encodes the OsCCD7 protein, which belongs to a family of non-heme iron enzymes linked to the production of volatiles, phytohormones and signals [43]. The decrease in three volatiles released from *d17* mutant plants may be an important reason for WBPH's preference.

*D17* is an essential synthetic gene of the SL pathway, which takes part in a number of defensive responses. When exposed to carbon dioxide, aphids exhibit higher fecundity on SL mutant plants than WT plants [44]. SL signaling mutant plants (*d14*) or biosynthesis mutant plants (*d17*) had increased susceptibility towards *Magnaporthe oryzae* [45]. A previous report also demonstrated that two rice mutants of SL biosynthesis and signaling, *d10* and *d14*, exhibited a series of transcriptional and metabolic changes, primarily in the lipid, flavonoid and terpenoid pathways, which play an essential role in plant immunity [46]. Additionally, the SL signaling pathways are also thought to interact with other plant hormones in the biological stress response [47,48]. For example, when *ccd7* plants (*Nicotiana attenuata*) are attacked by the specialist weevil (*Trichobaris mucorea*), the SL signaling pathway interacts with JA and auxin, attracting more weevils and producing larger larvae [29]. The SL biosynthesis and signaling pathway promotes root-knot nematode *Meloidogyne graminicola* infection by suppressing the JA signal, and the nematode infection can induce *D17* expression [49]. *Botrytis cinerea* and *Alternaria* alternate infection in tomato SL biosynthetic (*CCD8*) mutant lines is accompanied by a marked decrease in JA, SA and ABA content, indicating that SL mediates the defense against these pathogens by interacting with other defense-associated hormones [4]. Phytohormone pathways, including JA and SA, play a crucial role in plant defenses against herbivores [12–15]. For example, JAV1-JAZ8-WRKY51 complexes activate JA biosynthesis when plants are attacked by insects [50]. Compared to the *GRH2* near-isogenic line (TGRH11), the pyramided line (TGRH 29) carrying *GRH2* and *GRH4* exhibits strong resistance to green rice leafhoppers (GRH, *Nephotettix cincticeps* Uhler.), and *JAmyb* and *TPS* are significantly up-regulated in the TGRH 29 line following GRH infestation [51]. In this study, we found that the expression levels of the JA response genes *OsJAZ8* and *OsJAMyb* and the SA response genes *OsPR1a* and *OsWRKY62* were lower in the *d17* plants than the WT plants. These findings demonstrate that the *D17* mutation influences the JA and SA pathways when WBPHs attack, which makes it easier for WBPHs to hatch and survive on the *d17* mutant plants.

## 5. Conclusions

This study was conducted to investigate the difference in host selection behavior, number of eggs laid, and hatching and survival rates between *d17* and WT plants. We found that the *d17* plants were more attractive to WBPHs and more suitable for WBPH growth and reproduction than the WT plants. Additionally, at specific points, the content of hexanal, decanal and linalool was lower in the *d17* plants than the WT plants, and the expressions of JA- and SA-associated genes *OsJAZ8*, *OsJAMyb*, *OsPR1a* and *OsWRKY62* were significantly down-regulated when the plants were exposed to WBPHs. These findings demonstrated that the plants with the *D17* mutation were more vulnerable to attack by WBPHs. Since *D17* influences rice tillering and yield, our study serves as a new reference for future rice resistance breeding.

**Supplementary Materials:** The following supporting information can be downloaded at: https://www.mdpi.com/article/10.3390/agriculture13040842/s1, Table S1: Primers used for RT-qPCR of target gene.

**Author Contributions:** S.L. and W.D. conceived and designed the research; S.L., H.H., L.Q. and Q.G. performed experiments, collected and analyzed the data; S.L. and W.D. wrote the manuscript; Y.L. helped to revise the manuscript. All authors have read and agreed to the published version of the manuscript.

**Funding:** This research was funded by the National Key Research and Development Program of China (Grant No. 2021YFD1401100); the National Nature Science Foundation of China (Grant No. 32172506); and the Hunan Provincial Natural Science Foundation of China (Grant No. 2021JJ30317).

**Institutional Review Board Statement:** Not applicable.

**Data Availability Statement:** Data sharing is not applicable.

**Acknowledgments:** We thank Zeng, D.L. (China National Rice Research Institute) for providing the *d17* mutant plants.

**Conflicts of Interest:** The authors declare no conflict of interest.

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
