# Peer review of "Down-Regulation of Strigolactone Biosynthesis Gene D17 Alters the VOC Content and Increases Sogatella furcifera Infectivity in Rice"

_agriculture, doi:10.3390/agriculture13040842_

Round 1
Reviewer 1 Report
This study proports to examine the role of the D17 gene on WBPH and rice defenses to the insect. The authors documented the responses of WBPH to two wild type and SL-biosynthesis mutant rice and described volatiles associated with each plant. The authors also examined plant RNA for response gene activity after WBPH infestation. The paper raises some interesting possibilities and some aspects - particularly the graphs are very good. However, I have problems with the fundamentals of the study. Firstly, the authors give no mechanistic link between D17 and plant defenses against WBPH. Presumably, the D17 gene interacts with several other genes to result in dwarfism and high tillering. Other factors, for example some viral infections can cause the same effects on rice plants. The planthoppers responded to plants with a very different phenotype - plants that had more tillers, and therefore more space for oviposition, and that were more leafy. The authors therefore described a higher susceptibility of high-tillering, dwarf plants that is already largely recognized. This may be proximally related to D17, but not ultimately. The ultimate mechanisms are therefore important for this study. The authors then describe the role in defenses by some genes. The original bioassay was to infect the plants with 10 WBPH and examine RNA after different periods. But here again, we are not informed of possible pathways to defense. For example, high tillering plants might be affected by fewer hoppers per tiller, or they may be affected by more hoppers per biomass - this would affect the determination of responses, and in particular the titres. The experiment is therefore non-standardized and very prone to produce errors in statistical analyses. In effect, the study looks for too much from the D17 gene without thinking about ultimate mechanisms and possible plant-herbivore interactions. Alternate hypotheses are not considered at all - and not discounted by the methods. Finally, the statistical analyses are deficient. The authors conducted a series of block tests, but excluded the block effect. Time factors are also excluded - some results seem to be too strong, i.e., figure 3d error bars overlap considerably, but the factor is significant - are these standard errors. The analyses of volatiles requires a multivariate approach because each volatile is affected by the others. In short, the analyses need improvement. Finally the language of the manuscript requires substantial improvement - hungry = starved, spawn =oviposit, green revolution is used out of context. The references are also poorly selected and hardly respond to the information in the text.
Reviewer 2 Report
Manuscript ID: agriculture-2193815
Title: Strigolactones Biosynthesis Gene D17 Positively Regulate Defenses against Sogatella furcifera in Rice
Authors: Li et al.
General comments:
This manuscript is focused on the deciphering the role of Strigolactones Biosynthesis Gene D17 as a positive regulator of defense response against the insect Sogetella furcifera in rice. Certain sections of the manuscript are organized properly, and describe the relevant content appropriately. However, analysis, organization and presentation of data in results have several shortcomings, which need to be substantially improved. At certain places, coherence between the test and the data figures is missing. Certain analyses should be re-checked and if needed repeated for statistical significance. Some important concerns are also provided below, and also indicated in the PDF file of the manuscript.
1) The rationale of the study for analysis of D17 gene in insect resistance needs to be highlighted in a better manner
2) Materials and methods need improvement to provide proper details and previous publications at certain places. Language need to be improved at several places. RT-qPCR analysis and appropriate statistical analysis for various experiments needs to be provided in sufficient details.
3) The manuscript needs to be improved in both technical description and language style.
4) The discussion seems disconnected and losses connectivity with the introduction and background sections.
5) Statistical significance evident in the Figures and description is the results is mismatched and certain places.
Section specific comments
Title: The title of the manuscript looks appropriate in a way. However, check if it matches with the major results/discussion in study, and D17 indeed appears as a positive regulator of the defense against insect in rice. If needed, it may be modified.
Abstract: Abstract looks fine and minor suggestions are indicated for improvement.
Introduction: This section is appropriate; however rationale of study can be a little bit more emphatic. At certain places, minor changes are suggested for improvement (see PDF file for details). Some are briefly mentioned here
Lines 26-30: Description may need to be improved at certain places.
Material and methods: This section should be improved to provide sufficient details of methodologies used.
Line 63: Details of humidity conditions used should be indicated.
Line 67: When the experiment was carried out at 28-32°C, how and where 26°C was used for insect feeding.
Section 2.2: The details of Y-tube experimentation may be minimized by citing the previous publications. If not available, the selection of different parameters used for differentiating the insect behaviour on WT and mutant plants should be explained. A figure of Y-tube design in differential behaviour on insect may be included in supplementary data.
Section 2.3: Description needs to be re-written for better clarity of several statements.
Section 2.6: The details of RT-qPCR analysis, details of primers designed, and methods used for expression analysis should be elaborated a little bit. Statistical analysis methods should also be properly explained and used. At some places SE is indicated but in text SD is used. Also use of Chi-squared test or t-test should be clear and may be explained in authors response. The significance of results in certain places needs to be checked carefully.
Minor mistakes in the text should be rectified.
Results: Results section needs improvement due to concerns like, 1) content similar to materials and methods, which may be removed/minimized, 2) mis-match in gene names or figure in the text vis-à-vis figures, 3) mis-match of statistical significance in the figures and text description. Some of these concerns are indicated in the PDF file, however it should be ensured that the description/content at other places also should be carefully placed in appropriate sections of the manuscript.
Lines 136-138: This section can be part of Materials and Methods, If already there, repetition may be removed/minimized.
Lines 142-143: The statistical analysis of Figure 1B seems in consistent with the description and may be repeated. SD or SE used may be corrected. And use of CH-squared test may also be explained in Authors respone to comments. As Chi-squared test compares expected and observed outcomes, what is the expected outcome in these experiments is not clear. Will t-test to compare the data mean and SD not appropriate, here and why?
Line 169-160: The VOC content differences are not consistent at every time-point? Kindly check, and modify the statement if required.
Lines 161-163: Standard concentration terms may be used than V/V based description. Also What was the basis of same concentration of all the compounds? Some may work at higher/lower concentration. Any previous reports in support of used concentration may be indicated.
Line 167: In Figure 2C, check the labels of the Blue/Green plots? Seems in correct. Also Linalool data is apparently not from multiple analysis., so statistical assessment is not feasible. It may be rectified.
Lines 177-180: In Figure 3c, 3d and 3e the interpretation based on statistical significance indicated some inconsistency. There seems to be no significant difference between the hatchability, egg stage time on WT and d17 plants. The graph shows two stars (indicative of P <0.01). It must be clarified. See other plots also with mean, SD and P values? This may also be checked carefully for statistical significance. If mean and SD values are to compared for significance, at every time-point the data seems non-significant. The choice of tests (t-test and Chi-square) for statistical significance at some places needs to be re-assessed.
Lines 194-196: Is it possible to measure the levels of JA and SA first. Since sometimes the expression differences do not correspond to protein levels of phenotypic outcome.
Line 198-211: The concerns in the description about the basal levels (0h), up/down-regulation of genes, and statistical analysis, and missing figures must be addressed. Generalized statements of response based on 1-2 time points should be avoided.
Discussion: This section needs to be improved to enhance the integration of present study with the previous reports. Straightforward statements about role of D17 gene should be avoided, as it may need more in-depth analysis. OsWSKY26 Gene data is missing in the results, so its pattern may not be discussed (lines 221-222). Generalized statements about gene responses, when only 1-2 time-points are showing differences, should be avoided. Statements similar to results (lines 228-230) should be removed or description minimized. The VOCs levels are not completely turned down in d17 plants (see lines 235-236). Also at certain places in the discussion the role of VOCs or signalling genes is down played and involvement of silicon content or other traits is also discussed (lines 239-244). Also the statement indicating that the mutation d17 has caused down-regulation of insect resistance genes (lines 267-268) may need more evidences. Is there any study that shows what happens when D17 gene is over expressed? Then it would be appropriate to show its positive role in insect defense. So the basis of defense to insect is somewhat confusing and may need more experimental analysis.
Table 1 may be moved to the supplementary data
Figures: There are few minor concerns related to the Figures. See the PDF files for deatils
Statistical analysis data in certain figures should match the description of results in the text. e
Figure 2c: kindly recheck legend labels.
Figure 3: Check statistical significance data carefully.
Figure 4f is not discussed in the results and
Figure of OsWRKY26 is missing.

Round 2
Reviewer 2 Report
Manuscript ID: agriculture-2193815
Authors: Li et al.
General comments:
This revised version of the manuscript has addressed the concerns in an appropriate manner and now seems considerable improved. The following points may still need to be addressed
1. Title still not very emphatic, one suggestion: 'Down-regulation of Strigolactones Biosynthesis Gene D17 alters the VOC content and reduces Sogatella furcifera infectivity in Rice'. See if this seems appropriate.
2. The Y-tube figure can be part of the manuscript.
3. Figure 2c missing in the text. The x-axis title may not be 'Host selection preference' as there is not host plant used. Some appropriate axis title may be used, if applicable.
4. In Fig. 3e, 8-day time point seems insignificant. Only average data seems to have been provided in the response document, the SD (evident as error bars) is indicative of considerable overlap. May be re-checked and statements modify accordingly.
